# Chromatin remodeler Fft3 plays a dual role at blocked DNA replication forks

Anissia Ait-Saada[2], Olga Khorosjutina[1], Jiang Chen[1], Karol Kramarz[2], Vladimir Maksimov[1], J Peter Svensson[1], Sarah Lambert[2], Karl Ekwall[1]

Here, we investigate the function of fission yeast Fun30/Smarcad1 family of SNF2 ATPase-dependent chromatin remodeling enzymes in DNA damage repair. There are three Fun30 homologues in fission yeast, Fft1, Fft2, and Fft3. We find that only Fft3 has a function in DNA repair and it is needed for single-strand annealing of an induced double-strand break. Furthermore, we use an inducible replication fork barrier system to show that Fft3 has two distinct roles at blocked DNA replication forks. First, Fft3 is needed for the resection of nascent strands, and second, it is required to restart the blocked forks. The latter function is independent of its ATPase activity.

## Introduction

In eukaryotic chromosomes, the DNA is packaged into chromatin fiber structures to allow for compaction of DNA in the nucleus and proper chromosome segregation in mitosis and meiosis. The basic unit of the chromatin fiber is the nucleosome consisting of 146 bp of DNA wrapped around a protein structure of histones. Processes that need access to the DNA helix, for example, gene transcription, DNA replication, and repair are aided by nucleosome remodeling factors, which can disassemble or slide nucleosomes at a given genomic locus. SNF2 enzymes are ATP-dependent nucleosome remodeling factors with a conserved helicase-like domain (Flaus et al, 2006). Fun30 belongs to a subfamily of SNF2 enzymes with important roles in genome stability, gene regulation, and chromosome boundary function.

We and others have characterized the function of Fun30 homologues in fission yeast, *Schizosaccharomyces pombe*. There are three homologues called (Fission yeast Fun Thirty) Fft1, Fft2, and Fft3. Fft2 and Fft3 both have roles in regulation of expression and mobility of retrotransposable elements (Persson et al, 2016). Fft3 has functions at tRNA genes and LTR elements acting as chromosomal boundaries at centromeres and telomeres (Stralfors et al, 2011;

Steglich et al, 2015). Fft3 was recently implicated in transcription elongation by RNA polymerase II, where it is involved in disassembly and reassembly of nucleosomes to facilitate transcription (Lee et al, 2017). Another recent study found Fft3 in a genetic screen for factors needed for inheritance of heterochromatin at the silent mating-type loci. It was shown that Fft3 interacts with DNA replication factors (pol δ and ε) and is needed for proper DNA replication (Taneja et al, 2017). Moreover, Fft3 has also recently been identified by several high throughput genetic and proteomic screens in fission yeast. The first screen was monitoring silencing defects close to the domain boundary at the silent mating-type loci (Jahn et al, 2018). The second screen was identifying genes that interact with genes encoding Hamartin or Tuberin proteins involved in the human Tuberous sclerosis complex disorder, a benign tumor disease (Rayhan et al, 2018). Finally, Fft3 was identified by a proteomic screen for proteins bound to a meiotic recombination hotspot and was shown to be one of several chromatin regulators required for efficient recombination at the *ade6-M26* hotspot (Storey et al, 2018).

Both in budding yeast, *Saccharomyces cerevisiae*, and in mammalian cells, Fun30 homologues have been implicated in DNA double-strand break (DSB) repair, specifically in DNA end resection during homologous recombination (HR) (Chen et al, 2012; Costelloe et al, 2012; Eapen et al, 2012). The long-range 5′ to 3′ end resection, mediated by exonucleases such as Exo1, leads to long stretches of single-stranded DNA (ssDNA). Interestingly, a recent study found that these ssDNA structures may maintain association with histones, forming nucleosome-like structures, and Fun30 was effectively activated in vitro by such particles (Adkins et al, 2017). Short-range resection mediated by Exo1 and Fun30 was recently implicated in facilitating the mismatch DNA repair process (Goellner et al, 2018). Consistent with this notion, the Xenopus Fun30 homologue Smarcad1 was recently shown to facilitate nucleosome exclusion during mismatch repair (MMR) (Terui et al, 2018). Thus, Fun30 and its Smarcad1 homologue have established roles in resection during DNA repair processes.

Here, we have investigated the function of the three Fun30 homologues Fft1, Fft2, and Fft3 in *S. pombe* DNA damage repair. Two of the homologues, Fft1 and Fft2, do not seem to be involved in DNA repair. In contrast, the third homologue, Fft3, is important for

---

[1]Department of Biosciences and Nutrition, Karolinska Institutet, Huddinge, Sweden   [2]Institut Curie, Paris-Saclay University, Centre National de la Recherche Scientifique, Unités Mixtes de Recherche 3348, F-91405, Orsay, France

Correspondence: Sarah.Lambert@curie.fr; karl.ekwall@ki.se

DNA repair and cells lacking Fft3 are sensitive to several DNA-damaging drugs. A series of experiments have revealed a role for Fft3 in promoting single-strand annealing (SSA) and HR-mediated replication fork restart. We have uncovered a dual role for Fft3 at the stalled replication forks. Fork resection is dependent of the ATPase activity of Fft3, whereas the subsequent step of fork restart is facilitated by Fft3 but is independent of its ATPase activity.

## Results

### The Fft3Δ mutant has a defect in SSA

To test if any of the *S. pombe* Fun30 genes are involved in repair of DNA damage, we performed growth assays on plates containing the DNA-alkylating agent methyl methanesulfonate (MMS) that induces damaged replication forks. MMS alkylates guanine and adenine to cause mispairing and replication block. In the repair process, ssDNA breaks and gaps are produced, serving as a substrate for HR. *S. pombe* strains harboring gene knockouts for *fft1Δ*, *fft2Δ*, *fft3Δ*, and wild-type control were serially diluted and spotted onto YES plates with 0.005 and 0.01% MMS (Fig 1A). After 4 d of incubation at 30°C, only one of the mutants, *fft3Δ*, displayed increased MMS sensitivity as compared with wild-type control, in accordance with the recent report from Taneja et al, (2017).

To investigate the mechanistic role of Fft3 in DNA damage repair, we used an assay for resection of a single DSB (Watson et al, 2011). This system (HOcs-SSA) is based on the *MATα* HO-endonuclease cutting site placed into the *his3⁺* gene flanked by a disrupted *S. cerevisiae LEU2* marker gene ~5 kb away on each side (Fig 1B). Upon HO induction by addition of uracil to activate *urg1-HO*, DSBs are generated. The disrupted *LEU2* marker gene has a stretch of homologous sequence, allowing the SSA process of DSB repair to occur. Effective SSA results in a functional *LEU2* allele accompanied by the loss of the *his3⁺* gene. Induction of the DSB but failure to complete the SSA pathway will result in the loss of the *his3⁺* marker and *LEU2* gene. The *his3⁺* gene could also be lost by processing of the DSB through other repair pathways including nonhomologous end joining (NHEJ), as NHEJ will generally induce a frameshift. The *fft1Δ*, *fft2Δ*, and *fft3Δ* mutations were introduced in this model. As a positive control, we used a strain harboring a gene deletion for the Exo1 exonuclease required for the resection of DSBs and SSA products. After DSB induction, the colonies were allowed to form on nonselective media. To compensate for differences in growth rates, *fft1Δ* and *fft2Δ* mutants were grown for 5 d and, whereas the slower *fft3Δ* was grown for 7 d. After this incubation, the plates were replica-plated to the media lacking histidine and/or leucine and incubated for two additional days continuously. After this selection, the number of His⁻ Leu⁺, His⁻ Leu⁻, and His⁺ Leu⁻ colonies was quantified (Fig 1C). Cell viability after HO induction was interpreted as completed DSB repair. The only mutants that displayed decreased DSB repair were *fft3Δ* and *exo1Δ*. The *LEU2*-interspersed sequence of 16 His⁻ Leu⁻ colonies was sequenced (11 from *fft3Δ* and 5 from *exo1Δ*), expecting NHEJ products. However, NHEJ products were not found. Instead, all *fft3Δ* clones had the sequence expected

from SSA processing, but for unknown reason, these clones had not grown on the media lacking leucine. Possible explanations include that the breaks were repaired slower resulting in a delay in colony formation, or epigenetic silencing of the *LEU2* locus.

Next, we measured the induction of DSB at the *his3⁺* gene in wild-type, *fft2Δ*, and *fft3Δ* cells by qPCR (Fig S1). We observed an approximately twofold reduction of DSB in *fft3Δ* cells after 5 h that could contribute to a delay in the SSA process. To directly test the possibility of a delayed SSA repair, we then measured the appearance of the restored *LEU2* gene by qPCR at different time points after HO induction. To control for the observed different efficiencies of DSB induction, we normalized the data to evaluate only the cells where a DSB had occurred and followed the kinetics of repair by SSA (Fig 1D). We found that in *fft3Δ* cells, the appearance of SSA products was indeed delayed, whereas in *fft2Δ* cells, SSA products appeared with similar kinetics as in the wild type. Thus, no defect in DSB induction or SSA kinetics was detected in *fft2Δ*. In contrast, the *fft3Δ* mutant had a clear defect in both the DSB induction and the kinetics of repair by SSA. Hence, we conclude that unlike its paralogues Fft1 and Fft2, Fft3 plays a role in promoting SSA.

### The ATPase domain of Fft3 is needed for cell resistance to replication stress

Fft3 is an ATPase enzyme and the ATPase domain is essential for its nucleosome remodeling catalytic activity used for nucleosome remodeling. To test if the catalytic activity of Fft3 is needed for its function in DNA repair, we performed growth assays on plates containing a panel of DNA-damaging agents. This panel included MMS; camptothecin (CPT), an inhibitor of topoisomerase I inducing replication forks impediments; hydroxyurea (HU), an inhibitor of the ribonucleotide reductase inducing a global replication fork slow down and stalling; and bleomycin (Bleo), a DSB-inducing drug. *S. pombe* strains with epitope-tagged *fft3-myc*, the epitope-tagged ATPase-deficient allele *fft3-K418R-myc,* carrying a gene deletion *fft3Δ*, and a control strain deleted for the recombinase Rad51, *rad51Δ*, were serially diluted and spotted onto YES plates with different concentrations of the agents (Fig S2). The *fft3-K418R-myc* strain exhibited similar sensitivity to CPT and MMS than *fft3Δ* cells, indicating that the ATPase activity is required to promote cell resistance to replication stress. No increased sensitivity to Bleo was observed in cells lacking Fft3 or only its ATPase activity in contrast to *rad51Δ*. This indicates that Fft3, in spite of its role in DSB repair by SSA, is largely dispensable to promote survival after DSB induction. In contrast to a previous report (Taneja et al, 2017), we found that *fft3Δ* and *fft3-myc* cells were not sensitive to HU treatment, whereas *fft3-K418R-myc* cells showed a slightly higher sensitivity to HU treatment. These data reveal that, under specific circumstances, the lack of the ATPase activity of Fft3 can be more toxic than the lack of the protein itself, further supporting the importance of the ATPase activity of Fft3 to promote cell resistance to replication stress.

### Fft3 is needed for efficient DNA resection at blocked replication fork

To assess the mechanistic roles of Fft3 in the process of stressed replication forks, we have used the conditional *RTS1*-replication

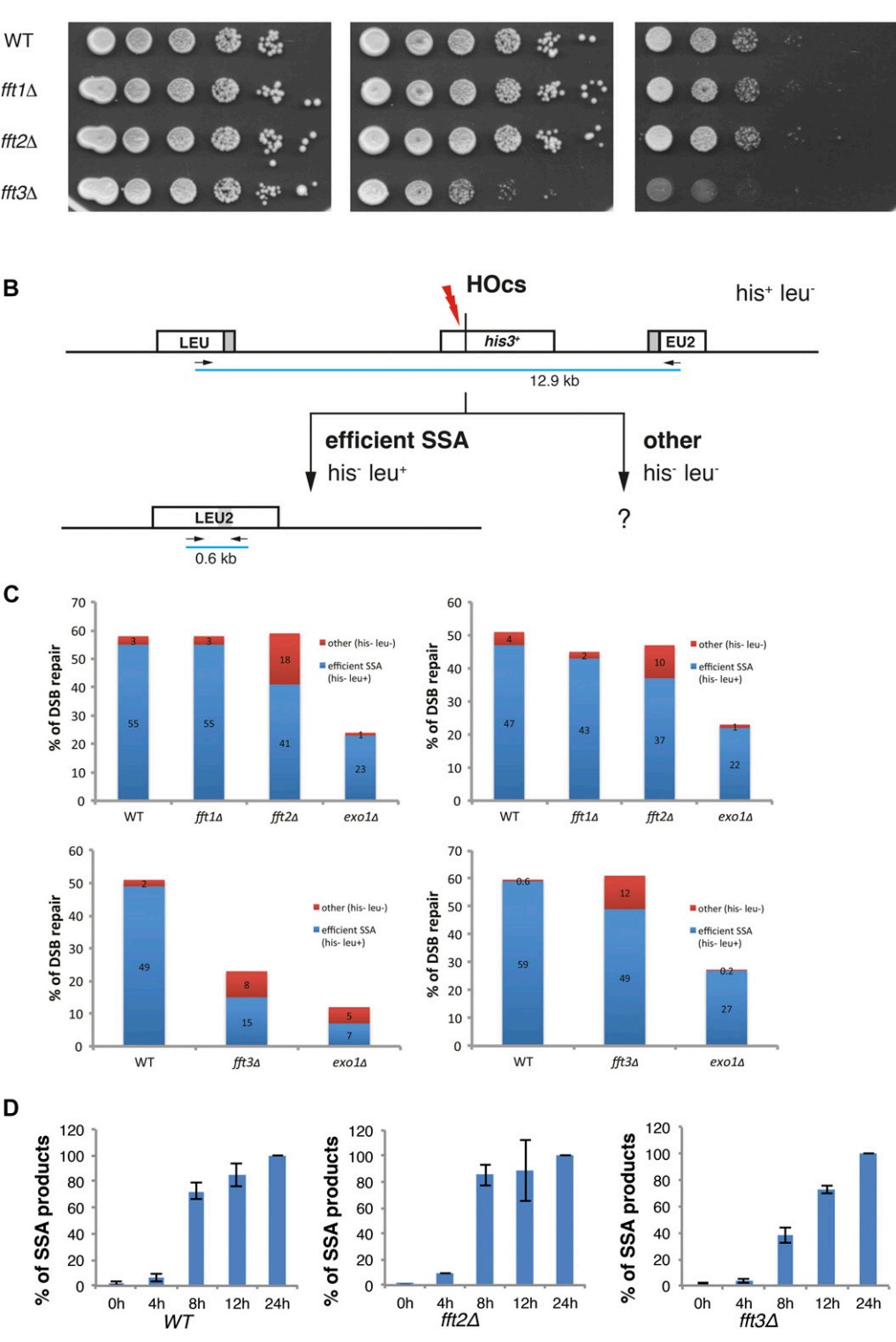

**Figure 1. Fft3 has a role in DNA damage repair and promotes SSA.**
**(A)** Spotting assays using DNA-damaging drugs. Photographs of YES plates after 4-d incubation at 30°C containing MMS at different concentrations (as indicated). The strain used were Hu0029 (WT), Hu2656 (*fft1Δ*), Hu1673 (*fft2Δ*), and Hu1309 (*fft3Δ*). **(B)** Schematic diagram of SSA assay from Watson et al (2011). His⁻ Leu⁺ colonies represent completed SSA events and His⁻ Leu⁻ colonies represent other repair events. Primers for quantitative PCR (arrows) and distances between them at the different genotypes are indicated. **(C)** Bar diagrams showing the percentage of DSB repair as viability after DSB induction in WT and the four mutants. Blue bars represent repair by SSA (interpreted by his⁻ leu⁺ phenotype) and red bars represent other repair (viable colonies with his⁻ leu⁻ phenotype). Processing after DSB induction on nonselective media was allowed for 5–7 d before phenotype testing (n = 2). **(D)** Bar diagrams showing the kinetics of SSA products after DSB induction, relative to the level at 24 h. The SSA product is estimated by quantitative PCR over the *LEU2* locus capturing the 44 bp product (n = 3, error bars show SD). For panels (B) and (C) the strains used were Hu2694, Hu2695, Hu2697, and Hu2698.

fork barrier (RFB) that allows a single replisome to be blocked in a polar manner at a specific locus (Fig 2A). The *RTS1*-RFB is encoded by the *RTS1* DNA sequence bound by the protein Rtf1, the expression of which is regulated by the thiamine-repressible *nmt41* promoter (Lambert et al, 2005). Upon expression of Rtf1, >90% of forks travelling in the main replication direction away from the centromere become arrested and dysfunctional at the *RTS1*-RFB (Lambert et al, 2010). Arrested forks are either rescued by a

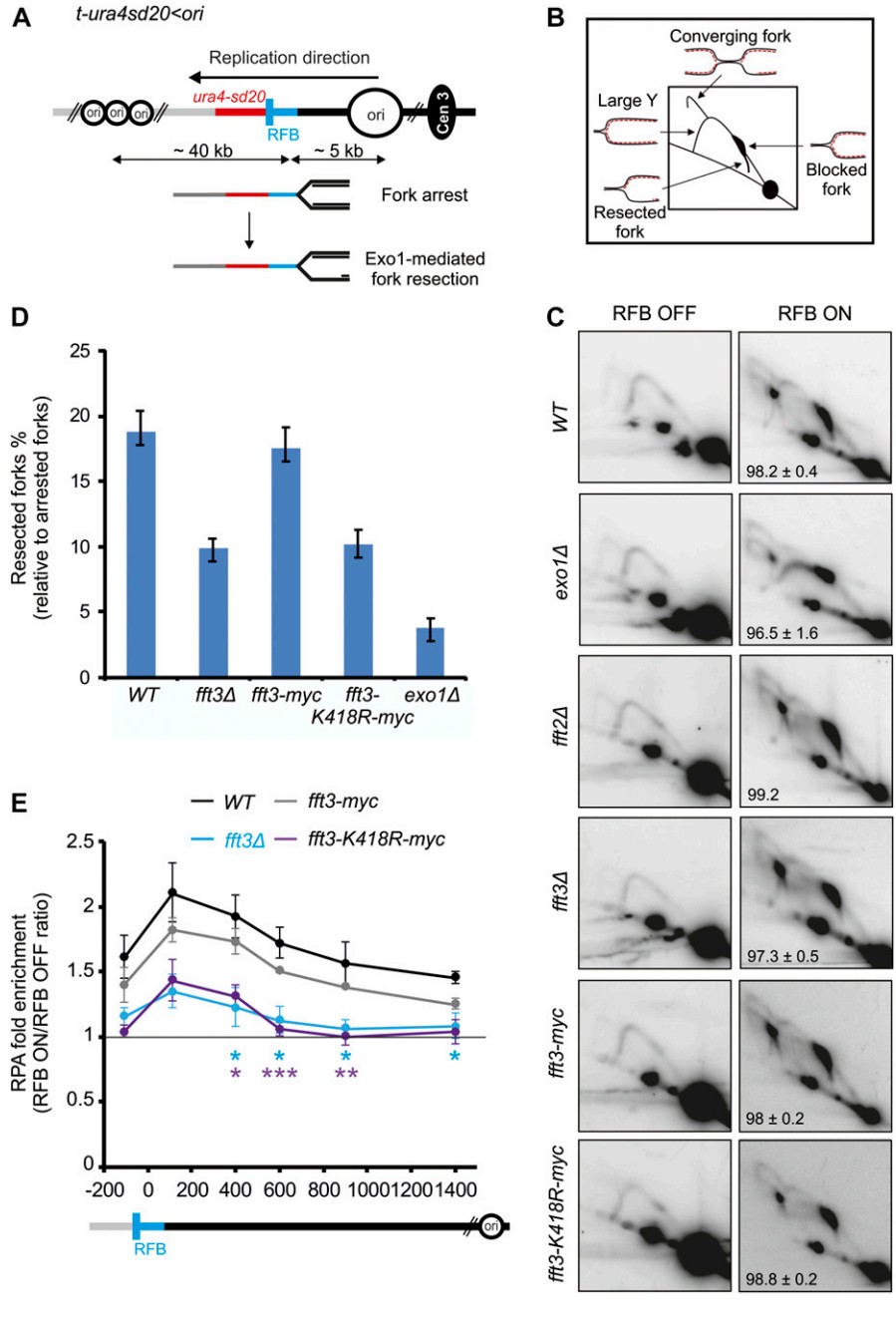

**Figure 2. Fft3 promotes DNA end resection at arrested forks through its ATPase activity.**
**(A)** Diagram of the *t-ura4sd20<ori* construct, on chromosome 3, containing a single *RTS1*-RFB (< and blue bars) blocking replication forks travelling in the main replication direction. Main replication origins (ori, black circles) located upstream and downstream of the *RTS1*-RFB are indicated with their respective distances from the RFB. When Rtf1 is expressed, >90% of forks emanating from the strong centromere-proximal replication origin, and moving towards the telomere, are blocked. HR-mediated fork restart is initiated through the generation of a 1-kb-sized ssDNA gap in an Exo1-dependent manner. **(B)** Scheme of replication intermediates (RIs) observed in a neutral–neutral 2DGE analysis of the *AseI* restriction fragment upon activation of the *RTS1*-RFB. **(C)** Representative RI analysis by 2DGE in the absence (RFB OFF) or presence of fork blockade (RFB ON) in indicated strains. A DNA fragment corresponding to *ura4* gene was used as probe. For each strain, numbers indicate the efficiency of the RFB; values are means of at least three independent experiments ± SD. **(D)** Quantification of % of fork undergoing resection (tail signals) relative to the number of blocked forks. Values are means of at least three independent experiments ± the 95% confidence interval (99% CI). In panels (C) and (D), the strains used were YC13 (*WT*), YC281 (*fft3Δ*), YC287 (*fft2Δ*), YC309 (*fft3-myc*), and YC313 (*fft3-K418R-myc*). **(E)** Binding of RPA (Ssb3-YFP) to the *RTS1*-RFB in indicated strains. ChIP-qPCR results are presented as an ON/OFF ratio for each strain. Upstream and downstream distances from the RFB are indicated in bp. Values are mean of four independent experiments ± SEM. *P* values were calculated using unpaired *t* test (\*\*\**P* ≤ 0.001; \*\**P* ≤ 0.01; \**P* ≤ 0.05). Data from *fft3Δ* (KK854 strain) were compared with wild-type (KK851 strain) (blue stars) and data from *fft3-K418R-myc* (KK857 strain) were compared with *fft3-myc* strain (KK860 strain) (purple starts).

converging fork or restarted by the HR pathway. HR-mediated fork restart occurs in 20 min through the generation of an ssDNA gap, which is subsequently coated by the Rad51 recombinase, with the help of its loader Rad52 (Tsang et al, 2014; Miyabe et al, 2015). After Rad51-mediated strand invasion of the parental duplex, the resumption of DNA synthesis occurs at the site of the arrested fork.

The generation of 1-kb-sized ssDNA gap at the active *RTS1*-RFB includes the resection of newly replicated strands by the nuclease Exo1 (Fig 2A) (Tsang et al, 2014, Ait Saada et al, 2017, Teixeira-Silva et al, 2017). The step of fork resection can be monitored by analyzing replication intermediates by two-dimensional gel electrophoresis (2DGE). We have previously reported a novel replication intermediate, emanating from blocked forks and descending toward the linear arc, corresponding to arrested forks in which newly replicated strands undergo Exo1-mediated end resection (Fig 2B) (Ait Saada et al, 2017). Consistent with this, the "tail signal" was nearly completely lost in *exo1Δ* cells (Fig 2C and D), as previously reported (Ait Saada et al, 2017). We found that fork resection was impaired in *fft3Δ* cells but not in *fft2Δ* cells. Quantification of the tail signal revealed a twofold reduction of the level of fork resection in *fft3Δ* cells compared with wild-type cells (Fig 2C and D). To test if the role of Fft3 in promoting fork resection requires its ATPase activity, we analyzed the ATPase-deficient allele Fft3-K418R (Steglich et al, 2015). We found a similar decrease in the level of resected forks as

observed in the null mutant (Fig 2C and D). Of note, the fusion of a myc-tag to Fft3 has no impact on fork resection, further supporting that Fft3-myc is functional as suggested by the insensitivity of the strain to genotoxic drugs (Fig S2). To further support the role of Fft3 and its ATPase activity in promoting end resection at arrested forks, we have used an alternative assay by analyzing the recruitment of Ssb3-YFP, one subunit of the ssDNA-binding protein RPA, to the *RST1*-RFB. The binding of RPA upstream from the *RTS1*-RFB is dependent on nucleases such as Mre11 and Exo1 and, thus, reflects the formation of ssDNA (Tsang et al, 2014). The recruitment of RPA upstream from the *RTS1*-RFB was significantly reduced in *fft3Δ* cells and in cells expressing Fft3-K148R (Fig 2E). This was particularly pronounced from 400 bp and more behind the arrested fork, indicating a less efficient long-range resection. We concluded that Fft3 and its chromatin remodeling activity promote the Exo1-mediated long-range resection of nascent strands at arrested forks.

### Chromatin-binding analysis at blocked DNA replication forks

To determine if Fft3 has a direct effect at blocked DNA replication forks, we performed ChIP of Fft3-myc–tagged cells and tested occupancy in the *RTS1*-RFB *ura4* L5 region (Fig 3A). We compared occupancy of Fft3-myc in RFB on and RFB off conditions using ± thiamine in the growth media. As a positive control, we used the valine tRNA gene known to be the site of Fft3 binding (Steglich et al, 2015). It was clear that binding occurred to L5 both in RFB on and off conditions at similar magnitude to valine tRNA genes (Fig 3B and C). In addition, the ATPase-deficient allele *fft3-K418R-myc* also showed binding to L5, regardless of the activity of the RFB, indicating that the fork resection defect observed in this mutant is not caused by an inability to bind the chromatin near the RFB. Nonetheless, the data obtained suggest that Fft3 is constitutively bound to the chromatin in the vicinity of the RFB regardless of its activity. To test this, we used another construct, *t-ura4sd20-ori*, devoid of the *RTS1* sequence (Fig 3D and E). We observed that Fft3-myc bound to L5 at similar magnitude to valine tRNA genes, irrespective of the media condition (with or without thiamine). We concluded that Fft3 is constitutively bound to the chromatin at the *ura4* locus, regardless of the activity of the RFB, preventing us to directly assess its specific recruitment to blocked replication forks.

### Fft3 is needed for efficient DNA replication restart at blocked replication fork

HR-mediated fork restart at the *RTS1*-RFB results in a restarted replisome, which is mechanistically distinct from a canonical replisome, with DNA polymerase δ synthetizing both strands (Miyabe et al, 2015). Restarted replisomes are error-prone, associated with a DNA synthesis intrinsically prone to replication slippage (RS) at regions of micro-homology (Iraqui et al, 2012; Mizuno et al, 2013) (Fig 4A). We have previously developed genetic assays to monitor the RS frequency occurring as a consequence of progression of the restarted replisome. The nonfunctional *ura4sd20* allele, containing a 20-nt duplication flanked by micro-homology, was integrated either upstream or downstream of the *RTS1*-RFB (Fig 4B). When the *ura4sd20* allele is replicated by a restarted replisome, the propensity of DNA polymerase to undergo RS allows the duplication to be deleted. In this

manner, a functional *ura4*⁺ gene is restored and Ura⁺ cells are generated. As a consequence of newly replicated strands undergoing end resection, restarted replisomes occasionally initiate upstream from the *RTS1*-RFB (Fig 4A). As control, a construct containing the reporter *ura4sd20* allele and devoid of the *RTS1*-RFB is used to monitor the frequency of spontaneous Ura⁺ cells in each genetic background (Fig 4B). Upon activation of the *RTS1*-RFB, the frequency of upstream RS was induced by ~2.6-fold in wild-type, *fft2Δ*, and *fft3-myc* strains, whereas no significant increase was observed in *fft3Δ* and *fft3-K418R-myc* strains (Fig 4C). These data are consistent with the step of fork resection being dependent on Fft3 and its ATPase activity, but not Fft2.

Upon activation of the *RTS1*-RFB, the frequency of downstream RS was induced by ~15-fold in wild-type and *fft2Δ* cells and by only approximately fourfold in *fft3Δ* cells (Fig 4D, top panel), indicating that only one-third of forks arrested at the *RTS1*-RFB are efficiently restarted in the absence of Fft3. Surprisingly, the induction of downstream RS in *fft3-K418R-myc* strain was similar to the one observed in wild-type cells (Fig 4D, bottom panel). This finding indicates that the lack of the ATPase activity does not impact the efficiency of HR-mediated fork restart. Collectively, these data establish a role for Fft3 in ensuring efficient HR-mediating fork restart, independently of its ATPase activity and its role in promoting fork resection.

# Discussion

### A role for Ff3 in DNA damage repair processes

Our work shows that of three Fun30 homologues, only Fft3 is implicated in the DNA damage repair in *S. pombe*. We show that Fft3 is required for proper SSA and processing of arrested replication forks. Both these processes depend on DNA resection requiring the ATPase domain and, hence, the nucleosome remodeling activity of Fft3. Similarly, Fun30 stimulates the long-range resection of DSBs but is dispensable for the initial resection (Chen et al, 2012; Costelloe et al, 2012; Bantele et al, 2017). The overexpression of Exo1 is sufficient to restore cell resistance of cells deleted for Fun30 to genotoxic drugs (Bi et al, 2015). Altogether, these data indicate an evolutionarily conserved function for Fft3 in promoting the Exo1-mediated long-range resection both at DSBs and blocked replication forks via its nucleosome remodeling activity. It is possible that Fft3 is also involved in MMR in *S. pombe* because Fft3 has been found to interact with the MMR protein Msh2 (Lee et al, 2017). In budding yeast, MMR depends on Fun30 activity and it is thought that the MMR process also requires DNA resection facilitated by Fun30 (Goellner et al, 2018).

### Several different functions of Fft3 in the nucleus

In addition to its role in DNA repair described here, Fft3 has previously been implicated in the chromosome domain organization of centromeres and telomeres by affecting LTR boundaries and tRNA boundaries (Stralfors et al, 2011; Steglich et al, 2015). A function of Fft3 in replication and propagation of heterochromatin has been shown by Taneja et al (2017) and Fft3 has been reported to be

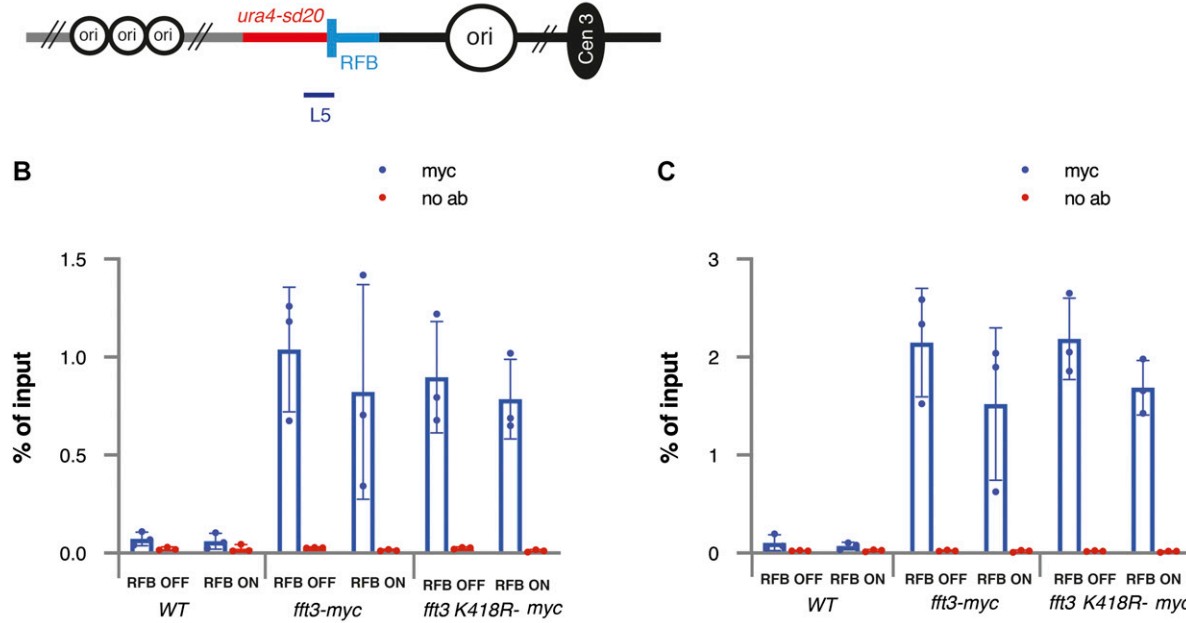

**A** t-ura4-sd20<ori

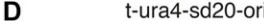

**D** t-ura4-sd20-ori

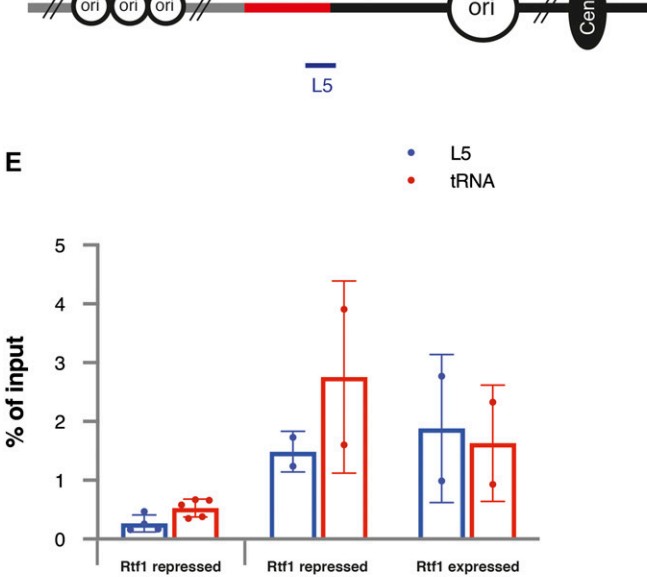

**Figure 3. Fft3 is associated with chromatin at the ura4 locus independently of the RFB.**
**(A)** Schematic diagram of *t-ura4sd20<ori* construct. The thick bar indicates position of PCR product at the *ura4* locus (L5) to detect Fft3-myc. **(B)** Fft3 recruitment to *RTS1*-RFB (L5) in indicated strains and conditions. Data from ChIP-qPCR of non-tagged wild-type, Fft3-myc, and Fft3-K418R-myc are shown. Enrichment is quantified as fraction of input. Error bars represent SD from three biological replicates. **(C)** Fft3 recruitment to valine tRNA genes. **(B)** Data and error bars as in (B). **(D)** Schematic diagram of *t-ura4sd20-ori* construct. **(E)** Fft3 recruitment to the *ura4* locus. The thick bar indicates position of PCR product at the *ura4* locus (L5) to detect Fft3-myc. In panels (B) and (C), the strains used were YC13 (*WT*), YC309 (*fft3-myc*), and YC313 (*fft3-K418R-myc*). In panel (E), strains used were YC13 (*WT*) and YC321 (*fft3-myc*).

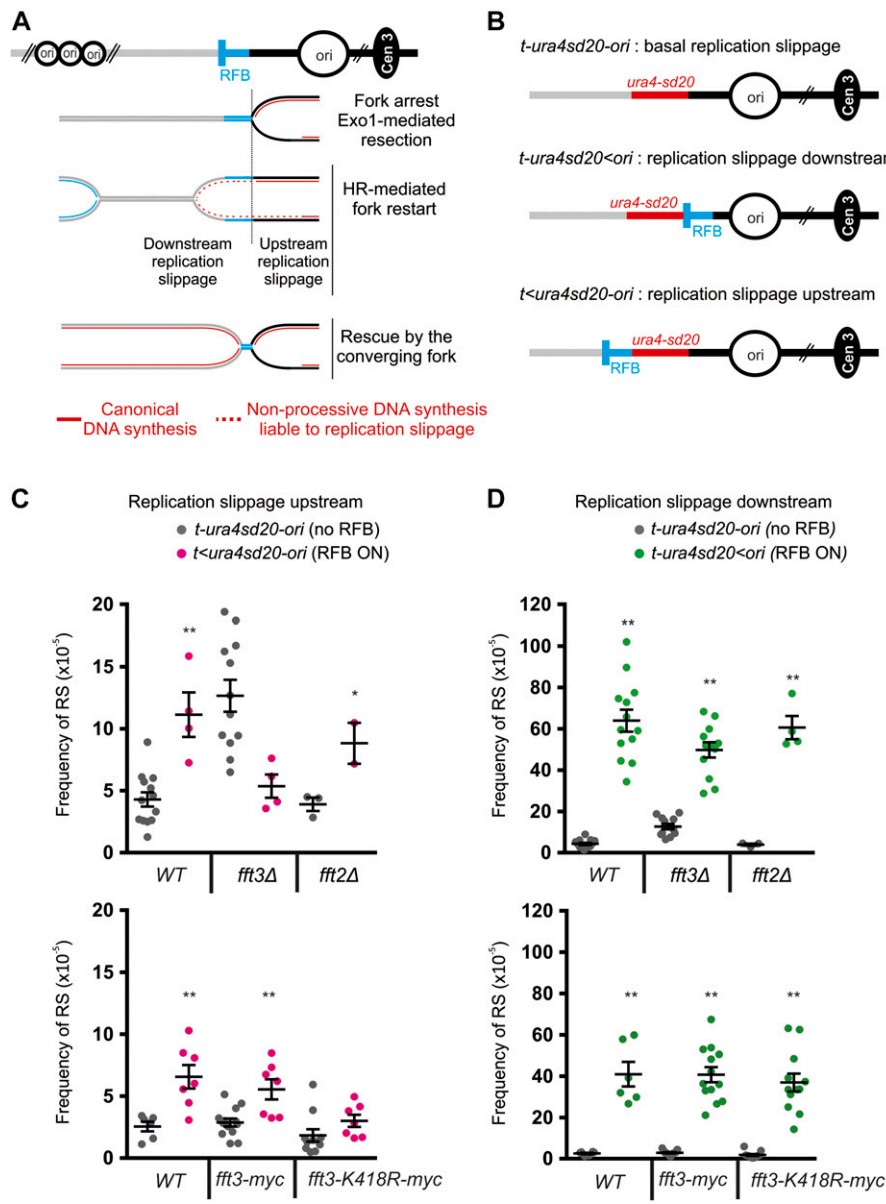

**Figure 4. Fft3 promotes replication fork restart independently of its ATPase activity.**
**(A)** Scheme of non-processive DNA synthesis associated to forks restarted at the *RTS1*-RFB (blue bar). Main replication origins (ori, black circles) located upstream and downstream of the *RTS1*-RFB are indicated. HR-mediated fork restart results in the progression of a restarted replisome associated to a non-processive DNA synthesis liable to RS (dotted red lines). Non-processive DNA synthesis can extend upstream of the *RTS1*-RFB as a consequence of fork resection. **(B)** Diagrams of constructs containing the reporter gene *ura4-sd20*: either not associated with the *RTS1*-RFB (top construct) or located downstream or upstream of the *RTS1*-RFB (middle and bottom constructs, respectively). The *ura4-sd20* allele contains a 20-nt duplication flanked by microhomology. When replicated by a restarted replisome, the associated non-processive DNA synthesis commits RS, allowing the duplication to be deleted and, thus, restoring a functional *ura4* gene and generating Ura+ cells. **(C)** Frequency of upstream RS (Ura+ cells) in indicated strains and conditions. The frequency of RS with the *t<ura4sd20-ori* was monitored upon Rtf1 expression (RFB ON). The construct *t-ura4sd20-ori*, devoid of RFB, was used as control to monitor the basal frequency of RS upon expression of Rtf1 (no RFB). Each dot represents one sample. Bars indicate mean values ± SEM. Statistical analysis was performed using *t* test (*$P < 0.04$, **$P < 0.004$). **(D)** Frequency of downstream RS (Ura+ cells) in indicated strains and conditions. The frequency of RS with the *t-ura4sd20<ori* was monitored upon Rtf1 expression (RFB ON). The construct *t-ura4sd20-ori*, devoid of RFB, was used as control to monitor the basal frequency of RS upon expression of Rtf1 (no RFB). Each dot represents one sample. Bars indicate mean values ± SEM. Statistical analysis was performed using *t* test (*$P < 0.04$, **$P < 0.004$). In panels (C) and (D), strains used were YC6, YC13, and YC21 for *WT*; YC280, YC281, and YC292 for *fft3Δ*; YC284, YC287, and YC294 for *fft2Δ*; YC321, YC309, and YC317 for *fft3-myc*; and YC329, YC313, and YC325 for *fft3-K418R-myc*.

required for silencing near the silent mating type heterochromatin domain boundary (Jahn et al, 2018). It is also involved in the control of gene expression (Lee et al, 2017). Finally, Fft3 is implicated in the repression of Tf2 expression and mobility (Persson et al, 2016). Thus, Fft3 is a key enzyme with multiple tasks in the nucleus.

## Mechanism of Fft3 function during DNA replication

Fft3 is an SNF2 helicase with a conserved ATPase domain belonging to the Fun30/Smarcad1 subfamily. This subfamily of SNF2 enzymes has previously been implicated both in nucleosome assembly and disassembly processes. In *S. pombe*, there is evidence for Fft3 carrying out both processes, for example, at Tf2 elements, a key regulatory nucleosome in the 5′ LTR is stabilized by Fft3. This stabilization leads to repression of Tf2 expression (Persson et al,

2016). In gene-coding regions, Fft3 contributes to nucleosome disassembly (Lee et al, 2017) and in the silent mating type region to nucleosome assembly, and roles in DNA replication, epigenetic inheritance, and suppression of nucleosome turnover have been demonstrated (Taneja et al, 2017). In other species, Fun30 and Smarcad1 play a role in nucleosome disassembly during DNA resection (Chen et al, 2012; Costelloe et al, 2012; Eapen et al, 2012). It is, therefore, likely that nucleosome disassembly is the relevant mechanism operating during DNA resection at blocked forks in fission yeast. This notion is consistent with the requirement of the ATPase domain of Fft3 for this function and the observation that single-stranded regions generated during resection have been shown as preferred substrate for Fun30 ATPase activity in vitro (Adkins et al, 2017). It is plausible that Fft3 maintains nucleosome-free regions promoting Exo1 activity during long-range resection

allowing for rescue of blocked forks by HR. It has been proposed that Fft3 precludes nucleosomes turnover to prevent the formation of natural fork obstacles, such as co-transcriptional R-loops, at highly transcribed genes, short repeats, and tRNAs (Taneja et al, 2017). In the absence of Fft3, replication defect occurs across various euchromatic loci. Here, we report novel functions by which Fft3 facilitates replication fork progression. First, Fft3 promotes the processing of arrested forks via its nucleosome remodeling activity. Second, Fft3 fine-tunes HR-mediated fork restart independently of its ATPase activity. Thus, Fft3 acts as a chromatin organizer to facilitate replication fork progression but also via an additional function unrelated to nucleosome remodeling. Regarding the ATPase-independent function of Fft3 in fork restart, a second mechanistic role of Fft3 is conceivable, perhaps involving the observed interactions with the DNA replication machinery (Taneja et al, 2017).

## Blocked forks and effects on Tf2 mobility

We previously showed that Fft3 is required to prevent mobility of the Tf2 class of retrotransposons in fission yeast (Persson et al, 2016). It is likely that this is a combined effect of reducing the expression of Tf2 and reducing the exposure of its possible insertion regions. Blocked DNA replication forks are known to be hot spots of Tf2 insertion (Jacobs et al, 2015). In *fft3Δ* cells, the number of stalled forks is expected to increase because the mechanism of fork processing and restart are impaired. It remains to be tested if Fun30 and Smarcad1 homologues have a conserved function in repression of transposon mobility.

## Genomic protection by Fft3 linked to its role in controlling nuclear organization?

Genome-wide mapping of Fft3-myc occupancy demonstrated enrichments at replication origins and several DNA repeat regions, including LTR elements and loci encoding tRNA, snoRNA, snRNA, and ncRNA (Steglich et al, 2015). DNA repeats are challenging for the DNA replication machinery and often cause impediments to fork progression. It is, therefore, tempting to speculate that Fft3 plays a general role in genome protection against DNA damage by its localization to these regions. A long-range DNA resection mechanism may be required at these loci to prevent unwanted recombination events between repeats when blocked forks are rescued by HR. Interestingly, some of these repeat elements also serve as chromatin domain boundaries (Allshire & Ekwall, 2015). We hypothesize that the protection of chromosomal regulatory domains and higher order chromatin domain structure by Fft3 could be linked to its role in homology-driven DNA damage repair.

# Materials and Methods

## *S. pombe* strains and growth conditions

The *S. pombe* strains used in this study are listed in Table 1. We used standard growth conditions and protocols for genetic experiments (Petersen & Russell, 2016; Ekwall & Thon, 2017). Strains carrying the *RTS1*-RFB were grown in the synthetic complete media EMM-Glutamate. The RFB was maintained inactive (RFB OFF) in the presence of 60 $\mu$M of thiamine in the medium. Activation of the *RTS1*-RFB (RFB ON) was achieved by removing thiamine from the medium and growing cells for 24 h. Sensitivity to genotoxic drugs was performed by spotting cells on media containing the appropriate drug.

## SSA assay

We used the SSA method for *S. pombe* cells described (Watson et al, 2011). Cells cultures were grown in liquid PMG+Leu medium at 30°C to logarithmic phase. Then, the cells were washed and resuspended at $5 \times 10^6$ cells/ml in PMG+Leu+His medium with Ura (to induce DSB) or without Ura (as non-DSB induced control). The cells were incubated at 30°C for 5 h. Immediately after this, the cells were counted and plated at 200–600 cells on PMG+Leu+His agar plates followed by incubation at 30°C for 5 d (*fft1Δ*, *fft2Δ*) or 7 d (*fft3Δ*) to recover and form colonies. Next, auxotrophy was tested as the plates were replica-printed onto PMG+His and PMG+Leu plates and incubated for another 2 d at 30°C. Finally, colonies were scored for growth on the different plates. Colonies relying on added Leu, but not His, for growth were scored as parental (no DSB induced or repaired without error), colonies requiring added His, but not Leu, were scored as "efficient SSA," and colonies requiring both added His and Leu were designated "other." The few colonies (<10) not needing either added His or Leu were not considered further. Colony numbers with parental phenotype (his⁺leu⁻) from the non-DSB induced control (without Ura) were used to determine the plating efficiency and the percentage of completed DNA repair. Also, non-DSB induced colonies with SSA phenotype (His⁻Leu⁺) were subtracted to account for "leakiness" of the *urg1*-system.

## qPCR

DNA was isolated and qPCR was performed with SYBR Master mix (Life technology) using the Applied Biosystems 7500 RT-PCR System. The primer sequences for the SSA product, covering the LEU2 locus were forward: 5′ GTG TTA GAC CTG AAC AAG GTT TAC, reverse: 5′ GCA AAG AGG CCA AGG ACG.

## DSB assay

The strains Hu2694 (WT), Hu2696 (*fft2Δ*), Hu2744 (*fft3Δ*), and Hu2745 (*fft3Δ*) were grown in liquid PMG+Leu medium, shaking overnight at 30°C, to the mid logarithmic phase (OD600 = 0, 5). The cells were washed with fresh PMG+Leu medium and resuspended at a concentration of five million cells per ml in PMG+Leu+His medium with Ura (to induce DSB) or without Ura (noninduced DSB control). The cells were incubated at 30°C for 5 h, and samples of 1 ml were taken at 0, 1, 2, 3, and 5 h followed by genomic DNA purification using the Thermo Fisher Scientific Yeast DNA Extraction Kit (Cat. no. 78870). qPCR was performed with purified genomic DNAs and primers to *his3-HO* and *act1+* genomic regions. The primer sequences for the *his3-HO* locus were forward: GATACAGTTCTCACATCACATCCG, reverse: CAGCGATAAGGCTGAAGTTCTAAG. The primer sequences for the *his3-HO* locus were forward: TCCAACCGTGAGAAGATGAC, reverse: TGTGGGTAACACCATCACCA. Upon DSB induction, cycling time values

**Table 1. List of *S. pombe* strains.**

| Strain name | Genotype | Source |
|---|---|---|
| Hu0029 | h- ade6-M210 leu1-32 ura4-D18 | Ekwall and Thon (2017) |
| Hu1309 | h+ fft3::kanMX ade6-M210 leu1-32 ura4-DS/E | This study |
| Hu1673 | h- fft2:: kanMX ade6-M210 leu1-32 ura4-D18 | This study |
| Hu2656 | h- ade6-M210 leu1-32 ura4-D18 fft1::KANMX | This study |
| Hu2694 | h- urg1::Purg1lox-HgD LEU-HOcs-his3-lambda-EU2, leu1-32 his3-D1 | This study |
| Hu2695 | h- urg1::Purg1lox-HgD LEU-HOcs-his3-lambda-EU2, leu1-32 his3-D1 fft1::KANMX | This study |
| Hu2696 | h- urg1::Purg1lox-HgD LEU-HOcs-his3-lambda-EU2, leu1-32 his3-D1 fft2::KANMX | This study |
| Hu2697 | h- urg1::Purg1lox-HgD LEU-HOcs-his3-lambda-EU2, leu1-32 his3-D1 fft3::KANMX | This study |
| Hu2744 | h- urg1::Purg1lox-HgD LEU-HOcs-his3-lambda-EU2, leu1-32 his3-D1 fft3::KANMX | This study |
| Hu2745 | h- urg1::Purg1lox-HgD LEU-HOcs-his3-lambda-EU2, leu1-32 his3-D1 fft3::KANMX | This study |
| Hu2698 | h- urg1::Purg1lox-HgD LEU-HOcs-his3-lambda-EU2, leu1-32 his3-D1 exo1::KANMX | This study |
| YC6 | h- rtf1:nmt41:sup35 ade6-704 leu1-32 t-ura4sd20-ori | Iraqui et al (2012) |
| YC13 | h- rtf1:nmt41:sup35 ade6-704 leu1-32 t-ura4sd20<ori | Iraqui et al (2012) |
| YC21 | h- rtf1:nmt41:sup35 ade6-704 leu1-32 t<ura4sd20-ori | Iraqui et al (2012) |
| YC281 | h- fft3:: HYGMX rtf1:nmt41:sup35 ade6-704 leu1-32 t-ura4sd20<ori | This study |
| YC292 | h- fft3:: HYGMX rtf1:nmt41:sup35 ade6-704 leu1-32 t<ura4sd20-ori | This study |
| YC280 | h- fft3:: HYGMX rtf1:nmt41:sup35 ade6-704 leu1-32 t-ura4sd20-ori | This study |
| YC287 | h+ fft2::KANMX rtf1:nmt41:sup35 ade6-704 leu1-32 t-ura4sd20<ori | This study |
| YC294 | h+ fft2:: KANMX rtf1:nmt41:sup35 ade6-704 leu1-32 t<ura4sd20-ori | This study |
| YC284 | h- fft2:: KANMX rtf1:nmt41:sup35 ade6-704 leu1-32 t-ura4sd20-ori | This study |
| YC309 | h- fft3-myc:HYGMX rtf1:nmt41:sup35 ade6-704 leu1-32 t-ura4sd20<ori | This study |
| YC317 | h- fft3-myc:HYGMX rtf1:nmt41:sup35 ade6-704 leu1-32 t<ura4sd20-ori | This study |
| YC321 | h- fft3-myc:HYGMX rtf1:nmt41:sup35 ade6-704 leu1-32 t-ura4sd20-ori | This study |
| YC313 | h- fft3-K418R-myc:HYGMX rtf1:nmt41:sup35 ade6-704 leu1-32 t-ura4sd20<ori | This study |
| YC325 | h- fft3K418R-myc:HYGMX rtf1:nmt41:sup35 ade6-704 leu1-32 t<ura4sd20-ori | This study |
| YC329 | h- fft3K418R-myc:HYGMX rtf1:nmt41:sup35 ade6-704 leu1-32 t-ura4sd20-ori | This study |
| KK851 | ssb3-YFP:NATMX rtf1:nmt41:sup35 t-ura4sd20<ori ade6-704 leu1-32 | This study |
| KK854 | fft3:: HYGMX ssb3-YFP:NATMX rtf1:nmt41:sup35 t-ura4sd20<ori ade6-704 leu1-32 | This study |
| KK857 | fft3K418R-myc:HYGMX ssb3-YFP:NAT rtf1:nmt41:sup35 t-ura4sd20<ori ade6-704 leu1-32 | This study |
| KK860 | fft3-myc:HYGMX ssb3-YFP:NAT rtf1:nmt41:sup35 t-ura4sd20<ori ade6-704 leu1-32 | This study |

for *his3-HO* region were increased because of DSB generation and subsequent loss of intact template for amplification compared with *act1+* gene sequence where no DSBs are normally generated. Therefore, the qPCR allowed us to measure DSB at *his3-HO* relative to *act1+*. Difference in cycling time (ΔCT) values were determined as CT(his3-HO) - CT(act1) and fold changes were calculated as (average E) to the power of ΔCT. Then, fold change differences were converted into percentages, given that ΔCT = 0 would represent 0% of DSB induction, and the maximal fold change difference value obtained was set as 100% of DSB induction. This approach allowed us to represent relative levels of DSB induction in the cells. Fold change values were used to calculate averages, SD and SEM values, and input data for statistical significance testing (*t* test).

## Analysis of replication intermediates by 2DGE

Replication intermediates were analyzed by 2DGE as described by Ait Saada et al (2017). 2.5 × 10$^9$ exponentially growing cells were harvested with 0.1% sodium azide and frozen EDTA (80 mM final concentration). The cells were cross-linked by adding trimethyl-psoralen (0.01 mg/ml, TMP, 3902-71-4; Sigma-Aldrich) to the cell suspensions, for 5 min in the dark. The cells were then exposed to UV-A (365 nm) for 90 s at a flow of 50 mW/cm$^2$. The cells were lysed with 0.625 mg/ml lysing enzyme (L1412; Sigma-Aldrich) and 0.5 mg/ml zymolyase 100T (120493-1; Amsbio) for 15 min at 37°C. Spheroplasts were then embedded into 1% low-melting agarose (InCert Agarose, Lonza) plugs and incubated overnight at 55°C in a digestion buffer containing 1 mg/ml of proteinase K (EU0090; Euromedex) and then washed and stored in TE (50 mM Tris and 10 mM EDTA) at 4°C. DNA digestion was performed with 30 units per plug of the restriction enzyme *Ase*I and equilibrated at 0.3 M NaCl. Replication intermediates were enriched using BND cellulose columns (B6385; Sigma-Aldrich) as described in Lambert et al (2010). Purified replication intermediates were then separated by bidimensional gel electrophoresis (0.35% agarose gel in TBE for the first dimension, 0.9% agarose gel-TBE supplemented with EtBr at 0.3 μg/ml). DNA was transferred to a nylon membrane in 10× SSC. Membranes were incubated with a $^{32}$P radio-labeled *ura4* probe, an RIs were detected using Phosphorimager software (Typhoon Trio) and quantified with ImageQuant TL.

## Chromatin immunoprecipitation of Fft3

DNA was immunoprecipitated as described earlier (Durand-Dubief & Ekwall, 2009) with following changes. Strains carrying the *RTS1-RFB* were cultured in supplemented EMMG media containing 60 μM thiamine. The cells were washed twice with water to remove thiamine and released into EMMG, either without (Rtf1 induced, RBF ON) or with (Rtf1 repressed, RBF OFF) 60 μM thiamine. After 24–25 h of RFB induction, 2 × 10$^8$ cells were fixed in 1% formaldehyde (252549; Sigma-Aldrich) for 30 min at room temperature with gentle agitation (Infors Multitron shaker; e120 rpm). To quench the cross-linking reaction, glycine was added to a final concentration of 125 mM. After 5-min incubation (RT, 120 rpm), the cells were collected by centrifugation (900*g*, 10 min, 4°C), washed twice with 25 ml of ice-cold PBS, and snap-frozen. The cells were resuspended in 400 μl of

cold CHIP lysis buffer CLB (50 mM Hepes-KOH, pH 7.5, 150 mM NaCl, 0.1% SDS, 1% Triton X-100, 0.1% sodium deoxycholate, 1 mM EDTA, and protease inhibitors [Complete Protease Inhibitor Cocktail EDTA–free, 11873580001; Roche]), transferred to the prechilled 2-ml skirted tube, containing 500 μl of zirconia/silica beads (11079105z; BioSpec products), and disrupted in a FastPrep-24 machine (116004500; MP Biomedicals) for two cycles (rate 6.5, 30 s on; 2 min off, 4°C). The crude lysates were sonicated (Bioruptor pico; Diagenode) for three cycles (30 s on; 60 s off, 4°C) and clarified by centrifugation (16,000*g*, 4°C, 10 min). The clarified chromatin extracts were transferred into fresh tubes. 5 μl of each chromatin extract was saved as input samples for subsequent quantification of the DNA enrichment. For each ChIP reaction, 100 μl of the extract was diluted with nine volumes of Binding Buffer B1 (as CLB, no SDS) to reduce SDS to a final concentration of 0.01%, with BSA added to 0.1% final concentration. The chromatin was immunoprecipitated using 2 μg of anti-Myc antibody (05–724, Upstate) for 2 h at 4°C, followed by incubation with 20 μl of pre-equilibrated Protein A/G magnetic beads (88802; Thermo Fisher Scientific) for 1 h. A sample without added antibody was prepared to assess unspecific DNA binding to the beads (later referred as no antibody [no ab] control sample). In addition, a parallel ChIP experiment was performed in the WT strain, lacking epitope tag, to evaluate efficiency and specificity of the anti-Myc antibody.

Beads were washed with 2 × 500 μl of cold B1 buffer, 1 × 500 μl B1/500 mM NaCl, 1 × 500 μl B2 (10 mM Tris-Cl, pH 7.5, 250 mM LiCl, 1 mM EDTA, 0.5% Na-deoxycholate, and 0.5% IGEPAL-630) and resuspended in 350 μl of TE (10 mM Tris-CL, pH 8, and 1 mM EDTA). The samples were transferred to fresh tubes, TE was removed, and 100 μl of TE, containing 0.1 μg/μl of RNaseA (10109169001; Roche), was added to each tube. The samples were incubated for 15 min at 37°C, treated with 0.5 μg/μl of Proteinase K (03 115 801 001; Roche) in 0.5% SDS for 1 h at 42°C, and subsequently incubated at 65°C overnight to reverse crosslinks. The input samples were mixed with 45 μl of TE containing RNaseA (0.2 μg/μl) and processed identically to the IP samples and the no ab samples.

The immunoprecipitated DNA was isolated using QIAquick PCR purification kit (28106; QIAGEN) according to the manufacturer's instructions. The DNA was eluted in 40 μl of elution buffer. For qPCR analysis, FastStart Universal SYBR Green Master (04 913 914 001; Roche) was used. 2 μl of the immunoprecipitated DNA were used as a template in total reaction volume of 12 μl. Gene-specific primers were used at a final concentration of 400 nM. The DNA quantity in the IP and input samples was determined using a standard curve method. The enrichment of the immunoprecipitated DNA was calculated relatively to the appurtenant input sample and presented as the percent of input DNA.

## Chromatin immunoprecipitation of RPA

Chromatin immunoprecipitation against RPA (ssb3-YFP) was performed as described in (Tsang et al, 2014) with following modifications. 200 ml of logarithmic culture (total of 2 × 10$^9$ cells) for each condition (*RTS1*-RFB OFF/ON) was divided into 2 × 100 ml aliquots and cross-linked with 10 mM dimethyl adipimidate (285625; Sigma-Aldrich) for 45 min and subsequently with 1% formaldehyde (F-8775; Sigma-Aldrich) for 15 min. Next, the cells from each 100-ml aliquot

**Table 2.** List of primers used for RPA ChIP-qPCR.

| Name | Distance (bp) from the *RTS1*-RFB position | Sequence (5'-3') |
|------|---------------------------|------------------|
| L5F | −110 | AGGGCATTAAGGCTTATTTACAGA |
| L5R | | TCACGTTTAATTTCAAACATCCA |
| L3F | 110 | TTTAAATCAAATCTTCCATGCG |
| L3R | | TGTACCCATGAGCAAACTGC |
| L400F | 450 | ATCTGACATGGCATTCCTCA |
| L400R | | GATGCCAGACCGTAATGACA |
| L600F | 600 | CCATTGACTAGGAGGACTTTGAG |
| L600R | | CCCTGGCGGTTGTAGTTAGT |
| L900F | 900 | AACGGTTGTAGAAGACGAGCA |
| L900R | | TGTAAGCACACCTTCAATGTATCA |
| L1400F | 1,400 | AACATCGGTGACCTCGTTCT |
| L1400R | | CTCTTCGCTCCAAGCGTTAT |
| II50F | Control locus on ChrII | CACCGCAGTTCTACGTATCCT |
| II50R | | CGATGTAACGGTATGCGGTA |

were frozen in liquid nitrogen and lysed by bead beating in 400 $\mu$l of lysis buffer (50 mM Hepes, pH 7.5, 1% Triton X-100, 0.1% Na-deoxycholate, 1 mM EDTA) with 1 mM PMSF, and complete EDTA-free protease inhibitor cocktail tablets (1873580; Roche). Chromatin sonication was performed using a Diagenode Bioruptor in a mode High, 10 cycles of 30 s ON and 30 s OFF in ice-cold water. Then sonicated chromatin fractions from each sample were pooled (400 + 400 $\mu$l), and immunoprecipitation overnight was performed as follows: 300 $\mu$l was incubated with anti-GFP antibody (A11122; Invitrogen) at 1:150 concentration, 300 $\mu$l was incubated with Normal Rabbit IgG antibody (#2729S; Cell Signaling Technology) at concentration 1:75 and 5 $\mu$l was preserved as an INPUT fraction. Next day, Protein G Dynabeads (10003D; Invitrogen) were added for 1 h and immunoprecipitated complexes and preserved INPUTs were de−cross-linked for 2 h at 65°C. DNA was purified with a QIAquick PCR purification kit (28104; QIAGEN) and eluted in 400 $\mu$l of water. qPCR (iQ SYBR Green Supermix, 1708882; Bio-Rad, primers listed in Table 2) was performed to determine the relative amounts of DNA (starting quantities based on standard curves for each pair of primers). RPA enrichment was calculated by dividing IP by INPUT values for specific (GFP) and unspecific (IgG) antibodies. Next, the values for unspecific IgG were subtracted and subsequently specific GFP signal was normalized over an internal control locus at chromosome II (II.50). The RPA enrichment was presented as ratio *RTS1*-RFB ON/OFF conditions.

## RS assay with *ura4-sd20* allele

RS using the *ura4-sd20* allele was performed as previously described (Iraqui et al, 2012). Ura+ cells were first counter-selected on 5-FOA plate. Single 5-FOA–resistant colonies were grown on uracil-containing plates with or without thiamine for 2 d at 30°C and then inoculated in uracil-containing EMM for 24 h. The cells were diluted and plated on YE plates (for survival counting) and on uracil-free plates containing thiamine to determine the reversion frequency. Colonies were counted after 5–7 d of incubation at 30°C. Statistics were performed using *t* test.

# Supplementary Information

# Acknowledgements

Work in the K Ekwall laboratory was supported by grants from the Swedish Cancer Society (Cancerfonden) and the Swedish Research Council (Veten-skapsrådet). Work in the S Lambert laboratory was supported by grants from the *Fondation pour la Recherche Médicale* "Equipe Fondation pour la Recherche Médicale DEQ20160334889," the Fondation ARC (l'Association pour la Recherche sur le Cancer), and the *Ligue* (*comité Essone*). A Ait-Saada was funded by the Fondation ARC.

## Author Contributions

A Ait-Saada: investigation and writing—review and editing.
O Khorosjutina: formal analysis, investigation, and writing—review and editing.
J Chen: investigation and writing—review and editing.
K Kramarz: investigation.
V Maksimov: investigation and writing—review and editing.
JP Svensson: conceptualization, formal analysis, and writing—review and editing.
S Lambert: conceptualization, supervision, funding acquisition, and writing—original draft, review, and editing.
K Ekwall: conceptualization, supervision, funding acquisition, and writing—original draft, review, and editing.

## Conflict of Interest Statement

The authors declare that they have no conflict of interest.

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
