## [Reviewer comments · Life Science Alliance]

Life Science Alliance

Chromatin remodeler Fft3 plays a dual role at blocked DNA replication forks

Anissia Ait-Saada, Olga Khorosjutina, Jiang Chen, Karol Kramarz, Vladimir Maksimov, J. Svensson, Sarah Lambert, and Karl Ekwall

DOI: <https://doi.org/10.26508/lsa.201900433>

Corresponding author(s): Karl Ekwall, Karolinska Institutet and Sarah Lambert, Institut Curie

Review Timeline:

Submission Date:	2019-05-20
Editorial Decision:	2019-06-07
Revision Received:	2019-09-02
Editorial Decision:	2019-09-19
Revision Received:	2019-09-23
Accepted:	2019-09-24

Scientific Editor: Andrea Leibfried

Transaction Report:

June 7, 2019

Re: Life Science Alliance manuscript #LSA-2019-00433-T

Dr. Karl Ekwall
Karolinska Institutet
Department of Biosciences and Nutrition
Novum
Stockholm S-141 83
Sweden

Dear Dr. Ekwall,

Thank you for submitting your manuscript entitled "Chromatin remodeler Fft3 plays a dual role at blocked DNA replication forks" to Life Science Alliance. The manuscript was assessed by expert reviewers, whose comments are appended to this letter.

As you will see, the reviewers appreciate your data and support publication of a revised manuscript in Life Science Alliance. We would thus like to invite you to submit a revised version to us, addressing the individual concerns raised. The reviewers provide constructive input and addressing the concerns seems straightforward, but please get in touch in case you would like to discuss individual points further with us.

Thank you for this interesting contribution to Life Science Alliance. We are looking forward to receiving your revised manuscript.

Sincerely,

Andrea Leibfried, PhD

Executive Editor
Life Science Alliance
Meyerhofstr. 1
69117 Heidelberg, Germany
t +49 6221 8891 502
e a.leibfried@life-science-alliance.org
www.life-science-alliance.org

B. MANUSCRIPT ORGANIZATION AND FORMATTING:

Reviewer #1 (Comments to the Authors (Required)):

The manuscript by Ait-Saada et al. documents the function of Fft3 in resection at DNA ends in fission yeast. While the function of its homologs in other fungi and in human is known, the role of Fft3 in fission yeast remained unknown. Here they show that Fft3 is important for resection at DNA break ends and at blocked replication forks. The data are convincing and important in the field of DNA repair. I have only few minor comments that need to be addressed:

All questions relate to Fig. 1B-D

1. Fft2 has more of his-leu- cells compared to WT. Is this difference significant. Do these also represent SSA event but Leu- or these are NHEJ.
2. His-Leu- colonies in fft3 are SSA products that remain Leu-. Are these stably Leu-? If these are stably Leu-, Sanger sequencing should be done to make sure there is no mutation at LEU2 sequence that occurred during SSA.
3. To make sure that SSA is slower in fft3 it would be good to show representative Southern blot to demonstrate that DSB induction is similar in fft3 and wild type cells. Slower SSA may result from slow resection or slow DSB induction in fft3.

Reviewer #2 (Comments to the Authors (Required)):

This is a nice little paper showing the role of the Fft3 SMARCAD homologue in fission yeast in replication fork stability and restart. The authors show that the related Fft1 and Fft2 proteins do not share this role. The experiments are well done and the data are generally convincing. The one concern I have is that the only evidence for delayed resection is using 2D gels. I would like to see an alternative assay for resection with more molecular detail to confirm this observation. Otherwise, I have no significant concerns .

Minor points: it would be helpful to have an English speaker go over the MS as there are minor grammatical issues.

Reviewer #3 (Comments to the Authors (Required)):

Anissia Ait-Saada, Olga Khorosjutina, Jiang Chen, J. Peter Svensson, Sarah Lambert, Karl Ekwall
Chromatin remodeler Fft3 plays a dual role at blocked DNA replication forks

Fission yeast Fun30/Smardc1 family of SNF2 ATPase is involved in histone turnover during transcription and DNA replication in vivo. Ait-Saada et al showed that only Fft3 could have a function in DNA repair utilizing growth assay on MMS plate, DSB repair assay, and RI analysis. Importantly, ATPase activity is required to promote cell resistance to replication stress, indicating that chromatin remodeling activity of Fft3 controls DNA repair process by SSA. Finally, Ait-Saada et al demonstrated a dual role for Fft3 at stalled replication forks. That is, ATPase of Fft3 is necessary for resection and is not necessary for HR-mediated fork restart.

Although we already know about Fun30's involvement in resection but we do not know the details about DNA repair process. Interestingly, Ait-Saada et al dissected the important point of Fun30 at blocked DNA replication forks. Overall, authors showed very interesting data and The results presented in the paper are rigorous. Before publication, I may add some suggestions below about their manuscript.

1. Figure 3: the corresponding dot plots have to be overlaid in the bar charts since the apparent error bars are too high.

2. Figure 4: It is interesting that K318R still showed comparable RS Frequency. Is there any possibility that other remodeler can be involved in this particular step? Or other Fun30 ortholog?

Point by point response to reviewer's comments

We thank all three reviewers for constructive criticisms that have helped to improve this manuscript. Below is our response to all the specific points.

Reviewer #1 (Comments to the Authors (Required)):

The manuscript by Ait-Saada et al. documents the function of Fft3 in resection at DNA ends in fission yeast. While the function of its homologs in other fungi and in human is known, the role of Fft3 in fission yeast remained unknown. Here they show that Fft3 is important for resection at DNA break ends and at blocked replication forks. The data are convincing and important in the field of DNA repair. I have only few minor comments that need to be addressed:

All questions relate to Fig. 1B-D

1. Fft2 has more of his-leu- cells compared to WT. Is this difference significant. Do these also represent SSA event but Leu- or these are NHEJ.

Answer: Yes the number of His-Leu- colonies is significantly higher for fft2D compared to wild type (unpaired t-test $p=0.036$). However these colonies were not sequenced so we can't say if they are SSA products or not.

2. His-Leu- colonies in fft3 are SSA products that remain Leu-. Are these stably Leu-? If these are stably Leu-, Sanger sequencing should be done to make sure there is no mutation at LEU2 sequence that occurred during SSA.

Answer: As mentioned on page 4 (second paragraph) sixteen His-Leu- colonies (11 from fft3D and 5 from exo1D) were sequenced and verified to have an intact LEU2+ gene. Therefore they can't be stably Leu- caused by a mutation in the LEU2+ gene.

3. To make sure that SSA is slower in fft3 it would be good to show representative Southern blot to demonstrate that DSB induction is similar in fft3 and wild type cells. Slower SSA may result from slow resection or slow DSB induction in fft3.

Answer: We have measured DSB in the mutants during a time course by QPCR. We observe lower DSB induction levels for fft3Δ strains compared to wild type and fft2Δ strains after 5 hours of DSB induction. Therefore, as pointed out by this reviewer, it is possible potentially slower DSB formation in his3-HO region could contribute to SSA products formation. However, due to design and calculations of SSA assay (Figure 1D), this seems unlikely. The SSA assay shows the kinetics of SSA product formation, relative to an arbitrarily chosen time point (24 hours in SSA assay). This also means only cells with induced DSB are being evaluated. For this reason it does not matter that much if at the start of an assay there are different amount of DSB events as long as sensitivity of detection method used

(qPCR) allows to follow product accumulation. Thus, we argue that there is no direct connection between slower SSA and slow DSB induction in *fft3Δ* and only slow resection is contributing into SSA product kinetics. Slow DSB induction in *fft3Δ* can result for instance from different expression levels of MATa HO-endonuclease which induces DSB in his3-HO region, due to function of *fft3* in transcription regulation.

We have included the DSB induction data as Supplementary Figure 2. See also text changes in results and methods sections.

Reviewer #2 (Comments to the Authors (Required)):

This is a nice little paper showing the role of the Fft3 SMARCAD homologue in fission yeast in replication fork stability and restart. The authors show that the related Fft1 and Fft2 proteins do not share this role. The experiments are well done and the data are generally convincing. The one concern I have is that the only evidence for delayed resection is using 2D gels. I would like to see an alternative assay for resection with more molecular detail to confirm this observation. Otherwise, I have no significant concerns .

Answer: As an alternative assay, we have analyzed the binding of RPA to the RFB by ChIP-qPCR, that reflects the formation of ssDNA (Tsang et al. J. Cell Science 2014). The recruitment of RPA upstream from the *RTS1*-RFB was significantly reduced in *fft3Δ* cells and in cells expressing Fft3-K148R. This was particularly pronounced from 400 bp and more behind the arrested fork, indicating a less efficient long-range resection. These data have been added as a new panel on figure 2 (Panel E). We believe that this alternative assay provides convincing molecular details to establish that Fft3 and its remodeling chromatin function are necessary to promote the resection of newly replicated strands at arrested forks.

Minor points: it would be helpful to have an English speaker go over the MS as there are minor grammatical issues. Answer: We have corrected grammatical errors.

Reviewer #3 (Comments to the Authors (Required)):

Anissia Ait-Saada, Olga Khorosjutina, Jiang Chen, J. Peter Svensson, Sarah Lambert, Karl Ekwall
Chromatin remodeler Fft3 plays a dual role at blocked DNA replication forks

Fission yeast Fun30/Smardc1 family of SNF2 ATPase is involved in histone turnover during transcription and DNA replication in vivo. Ait-Saada et al showed that only Fft3 could have a function in DNA repair utilizing growth assay on MMS plate, DSB repair assay, and RI analysis. Importantly, ATPase activity is required to promote cell resistance to replication stress, indicating that chromatin remodeling activity of Fft3 controls DNA repair process by SSA. Finally, Ait-Saada et al demonstrated a dual role for Fft3 at stalled replication

forks. That is, ATPase of Fft3 is necessary for resection and is not necessary for HR-mediated fork restart.

Although we already know about Fun30's involvement in resection but we do not know the details about DNA repair process. Interestingly, Ait-Saada et al dissected the important point of Fun30 at blocked DNA replication forks. Overall, authors showed very interesting data and The results presented in the paper are rigorous. Before publication, I may add some suggestions below about their manuscript.

1. Figure 3: the corresponding dot plots have to be overlaid in the bar charts since the apparent error bars are too high.

Answer: A new version of Figure 3 has been produced with dot plots as requested.

2. Figure 4: It is interesting that K318R still showed comparable RS Frequency. Is there any possibility that other remodeler can be involved in this particular step? Or other Fun30 ortholog?

Answer: We have tested the role of other Fun30 orthologues. Combining the deletion of *fft1* with *fft3Δ* or *fft3-K418R* mutation did not increase the cell sensitivity to MMS and did not impact the RS frequency. These data exclude a role for Fft1 in promoting fork-restart in the absence of the chromatin remodeling activity of Fft3 (see figure below).

Combining the deletion of *fft2* with *fft3Δ* or *fft3-K418R* mutation resulted in a severe growth defect that prevents us to apply the RS assay in these strains (see figure below). Also, this synthetic sickness may indicate a role for Fft2 in the absence of Fft3 and its ATPase activity, we cannot further test this hypothesis. Since the data obtained are negative or not fully conclusive, we did not add them to the current manuscript but we can provide them if the reviewer wish them to be included. We did not investigate the role of other chromatin remodeler in the time-frame of this reviewing.

Figure 1: Figure for reviewer 3.

Top panel: Serial 10-fold dilutions of indicated strains on indicated media.
 Bottom panel: Frequency of downstream Replication slippage in indicated strains in RFB OFF (OFF, blue bars) and RFB ON (ON, orange bars).

September 19, 2019

RE: Life Science Alliance Manuscript #LSA-2019-00433-TR

Dr. Karl Ekwall
Karolinska Institutet
Department of Biosciences and Nutrition
Novum
Stockholm S-141 83
Sweden

Dear Dr. Ekwall,

Thank you for submitting your revised manuscript entitled "Chromatin remodeler Fft3 plays a dual role at blocked DNA replication forks". We would be happy to publish your paper in Life Science Alliance pending final minor revisions, mainly necessary to meet our formatting guidelines:

- Please address the remaining comment of reviewer #1
- Please upload the manuscript file in docx format
- Please add a callout in the manuscript text to figure 3E
- I noticed that the % of DSB repair in figure 1C is based on two biological replicates. It would be good to show the individual bar graphs next to each other instead of the average.

A. FINAL FILES:

-- Summary blurb (enter in submission system): A short text summarizing in a single sentence the study (max. 200 characters including spaces). This text is used in conjunction with the titles of

papers, hence should be informative and complementary to the title. It should describe the context and significance of the findings for a general readership; it should be written in the present tense and refer to the work in the third person. Author names should not be mentioned.

B. MANUSCRIPT ORGANIZATION AND FORMATTING:

Sincerely,

Reviewer #1 (Comments to the Authors (Required)):

The revised manuscript is suitable for publication however the questions were not completely addressed. This reviewer is convinced that Fft3 promotes resection. This is the main message of the work. Somewhat side observation with SSA products that lead to Leu⁻ colonies could be explained better. The authors state in response to my question #2 (pasted below) that Leu⁻ colonies cannot be stably Leu⁻. How was it tested? Why is it not stated in the main manuscript? Sequencing is not the test for Leu2 expression. Was the phenotype tested? Re-streaked colonies were Leu⁺?

2. His-Leu⁻ colonies in fft3 are SSA products that remain Leu⁻. Are these stably Leu⁻? If these are stably Leu⁻, Sanger sequencing should be done to make sure there is no mutation at LEU2 sequence that occurred during SSA. Answer: As mentioned on page 4 (second paragraph) sixteen His-Leu⁻ colonies (11 from fft3D and 5 from exo1D) were sequenced and verified to have an intact LEU2⁺ gene. Therefore they can't be stably Leu⁻ caused by a mutation in the LEU2⁺ gene.

Point by point response to Reviewer's and Editor's comments

We thank reviewer #1 and the Editor for the additional comments that has helped to further improve this manuscript. Below is our response.

Reviewer #1 (Comments to the Authors (Required)): The revised manuscript is suitable for publication however the questions were not completely addressed. This reviewer is convinced that Fft3 promotes resection. This is the main message of the work.

Somewhat side observation with SSA products that lead to Leu- colonies could be explained better. The authors state in response to my question #2 (pasted below) that Leu- colonies cannot be stably Leu-. How was it tested? Why is this not stated in the main manuscript? Sequencing is not the test for Leu2 expression. Was the phenotype tested? Re-streaked colonies were Leu+?

We verified that the colonies were genetically *LEU2+* by sequencing, however as the reviewer points out, we did not test expression of the protein (by re-streaking or other phenotyping methods). As we cannot rule out that the protein levels were diminished after DNA repair, we have made the following modifications to the manuscript:

"A possible explanation would be that the breaks were repaired slower, resulting in a delay in colony formation." (page 4) has been changed to "Possible explanations include that the breaks were repaired slower resulting in a delay in colony formation, or epigenetic silencing of the *LEU2* locus."

Comment from the Editor

- Please add a callout in the manuscript text to figure 3E
Has been added on page 7
- I noticed that the % of DSB repair in figure 1C is based on two biological replicates. It would be good to show the individual bar graphs next to each other instead of the average.
Figure 1 has been changed so that individual bar graphs are shown for the biological replicates in Figure 1C

September 24, 2019

RE: Life Science Alliance Manuscript #LSA-2019-00433-TRR

Dr. Karl Ekwall
Karolinska Institutet
Department of Biosciences and Nutrition
Neo building
Stockholm S-141 83
Sweden

Dear Dr. Ekwall,

Thank you for submitting your Research Article entitled "Chromatin remodeler Fft3 plays a dual role at blocked DNA replication forks". I appreciate the introduced changes and it is a pleasure to let you know that your manuscript is now accepted for publication in Life Science Alliance. Congratulations on this interesting work.

DISTRIBUTION OF MATERIALS:

Again, congratulations on a very nice paper. I hope you found the review process to be constructive and are pleased with how the manuscript was handled editorially. We look forward to future exciting submissions from your lab.